



# A novel multinuclear solid state NMR approach for the characterization of kidney stones

César Leroy[1,4], Laure Bonhomme-Coury[1], Christel Gervais[1], Frederik Tielens[1,7], Florence Babonneau[1], Michel Daudon[2], Dominique Bazin[3], Emmanuel Letavernier,[2] Danielle Laurencin[4], Dinu Iuga[5], John V. Hanna[5], Mark E. Smith[6], Christian Bonhomme[1]

[1]Sorbonne Université, CNRS, Laboratoire Chimie de la Matière Condensée de Paris, LCMCP, F-75005 Paris, France.
[2]AP-HP, Hôpital Tenon, Explorations Fonctionnelles Multidisciplinaires et INSERM UMRS 1155, Sorbonne Université, Hôpital Tenon, Paris, France.
[3]Institut de Chimie Physique, UMR CNRS 8000, Bâtiment 350, Université Paris Saclay, 91405 Orsay cedex, France & Laboratoire de Physique des Solides, UMR CNRS 8502, Bâtiment 510, Université Paris-Sud, 91405 Orsay cedex, France
[4]ICGM, Univ Montpellier, CNRS, ENSCM, Montpellier, France.
[5]Department of Physics, University of Warwick, Gibbet Hill Road, Coventry CV4 7AL, United Kingdom.
[6]Department of Chemistry, University of Southampton, Southampton SO17 1BJ, United Kingdom.
[7]General Chemistry (ALGC) – Materials Modelling Group, Vrije Universiteit Brussel (Free University Brussels – VUB), Pleinlaan 2, 1050 Brussel, Belgium.

*Correspondence to*: Christian Bonhomme (christian.bonhomme@upmc.fr)

This article is dedicated to Geoffrey Bodenhausen on the occasion of his 70th Birthday.

**Abstract.** The spectroscopic study of pathological calcifications (including kidney stones) is extremely rich and helps to improve the understanding of the physical and chemical processes associated with their formation. While FTIR imaging and optical/electron microscopies are routine techniques in hospitals, there has been a dearth of solid state NMR studies introduced into this area of medical research, probably due to the scarcity of this analytical technique in hospital facilities. This work introduces effective multinuclear and multi-dimensional solid state NMR methodologies to study the complex chemical and structural properties characterising kidney stone composition. As a basis for comparison three hydrates (n = 1, 2 and 3) of calcium oxalate are examined along with nine representative kidney stones. The multinuclear MAS NMR approach adopted investigates the $^{1}H$, $^{13}C$, $^{31}P$ and $^{43}Ca$ nuclei, with the $^{1}H$ and $^{13}C$ MAS NMR data able to be readily deconvoluted into the constituent elements associated with the different oxalates and organics present. For the first time, the full interpretation of highly resolved $^{1}H$ NMR spectra is presented for the three hydrates, based on structure and local dynamics. The corresponding $^{31}P$ MAS NMR data indicates the presence of low-level inorganic phosphate species, however the complexity of these data make the precise identification of the phases difficult to assign. This work provides physicians, urologists and nephrologists with additional avenues of spectroscopic investigation to interrogate this complex medical dilemma that requires real multi-technique approaches to generate effective outcomes.





## 1 Introduction

Kidney stones (KS) are a major health problem in industrialized countries. For example, the medical costs associated with the treatment of nephrolithiasis in France exceeds 800 million € annually. The study of KS is presently at the heart of a concerted multi-disciplinary axis of research involving physicians, physical chemists and spectroscopists (Bazin et al., 2016). Nevertheless, the nucleation and growth of KS remains largely unknown and the associated mechanism is based mainly on assumption and incomplete evidence; hence, more thorough and wide-ranging structural investigations are still required

(Sherer et al., 2018; Bazin et al., 2020). The growth of KS is clearly a multi-factorial problem, with their chemical composition and morphology presenting considerable variability due to the extreme complexity of the *in vivo* reaction media in which they are formed. The resultant biological materials exhibit very different characteristics as they can emanate from wide-ranging pathological scenarios including bacterial infection, genetic predispositons, mellitus diabetes and bowel diseases (Bazin et al., 2012). Hence, KS can be considered as real examples of hybrid organic-inorganic nanocomposite materials.


The main mineral components comprising hydrated calcium oxalates are the monohydrate $CaC_2O_4 \cdot H_2O$ (whewellite, COM) and dihydrate $CaC_2O_4 \cdot 2H_2O$ (weddellite, COD) species, although amorphous calcium oxalate can also be observed (Gehl et al., 2015; Ruiz-Agudo et al., 2017). The trihydrate form, $CaC_2O_4 \cdot 3H_2O$ (caoxite, COT) is almost never observed *in vivo* but can be synthesized in aqueous solution. COD is characterized by a zeolitic structure exhibiting a true structural challenge. It is

considered as one of the very few *natural* MOFs (Metal Organic Frameworks) (Huskic, 2016; Dazem et al., 2019) and its chemical formula is better represented by $CaC_2O_4 \cdot (2+x)H_2O$ ($x \leq 0.5$) (Petit et al., 2018). "Structural" and "*zeolitic*" water molecules are therefore distinguished. Calcium phosphates and other mineral phases can be detected as well in KS: hydroxyapatite ($Ca_{10}(PO_4)_6(OH)_2$) which may be partially carbonated, brushite ($CaHPO_4 \cdot 2H_2O$) or struvite ($NH_4MgPO_4 \cdot 6H_2O$) (Gardner et al., 2021). The organic components (from few % to a major fraction) include e.g.: proteins

(collagen among them), uric acid, lipids, triglycerides, etc. The nature of the organic-inorganic interfaces remain largely unknown to date. This chemical and structural complexity at several scales requires the use of a wide variety of characterization methods. Recently, elaborate experiments took advantage of the last development in TEM (Transmission Electron Microscopy) (Gay et al., 2020) and of synchrotron radiation (Bazin et al., 2012). In hospitals, optical microscopy, FTIR, FTIR microscopy, SEM (Scanning Electron Microscopy) and X-ray diffraction are used in routine mode. Curiously, solid state NMR has been

used very rarely in the context of KS (and other pathological calcifications) apart from sparse [13]C and [31]P studies (Bak et al., 2000; Jayalakshmi et al., 2009; Reid et al., 2011; Reid et al., 2013; Li et al., 2016; Dessombz et al., 2016), unlike other human hard tissues such as bones and teeth. This is probably due to the fact that solid state NMR instruments are not widely available in hospital settings. It is also stressed that some KS are small so that the intrinsic lack of sensitivity associated to NMR may be a drawback. Other nuclei such as [1]H and [43]Ca can act as potential NMR targets. However, [1]H solid state NMR remains a

rather specialized technique because of the relative inefficiency of Magic Angle Spinning (MAS) in producing really high





resolution data from most systems. $^{43}$Ca (I = 7/2) is particularly insensitive (as a result of its extremely low natural abundance, ~ 0.14%, and low γ, −1.8028.10$^7$ rad.s$^{-1}$.T$^{-1}$, 57.2 MHz at 20 T).

In this work, a comprehensive multinuclear solid state NMR approach is presented facilitating the detailed structural analysis
of KS and the related synthetic hydrated calcium oxalate phases (COM, COD, COT) associated with their composition. The synthetic phases were obtained by carefully controlling the precipitation of calcium salts in aqueous solutions as described below in section 7 (Leroy, 2016). Nine KS were studied systematically, some of them exhibiting similar NMR fingerprints. The spectra of five of them (KS1 → KS5) are presented here. They come from the KS collection of Tenon hospital (Paris, France), led by Dr M. Daudon (the collection counts tens of thousands of samples from all origins exhibiting the largest variety
of size, chemical composition and morphology worldwide). Our main goal here is to reach out to the physician's community and more specifically nephrologists, urologists and biologists. NMR methods are presented at moderate to high magnetic field (*i.e.* 7.0 to 16.4 T MHz) in order to make them much more widely accessible. Occasionally, further developments at ultra-high magnetic field (up to 35.2 T) are proposed to the user. Particular emphasis is placed on high-resolution $^1$H MAS NMR with homonuclear decoupling and the complete interpretation of spectra based on structural data, and $^{43}$Ca MAS NMR. To the best
of our knowledge, these nuclei have never been used as spectroscopic probes for KS studies (apart for a unique $^{43}$Ca MAS NMR study by Bowers and Kirkpatrick, 2011). A complete experimental protocol is then presented for the reconstruction of $^{13}$C NMR spectra including organic/inorganic and/or rigid/mobile components. Finally, the intriguing role of phosphates in KS is partially deciphered by 2D $^1$H–$^{31}$P HETCOR MAS NMR experiments despite the low phosphate content in KS.

## 2 Quick and reliable assignment of hydrated calcium oxalate and organic phases by $^1$H high resolution solid state NMR
experiments

### 2.1 CRAMPS (Combined Rotation And Multiple Pulses Spectroscopy) approach

In terms of NMR sensitivity, $^1$H greatly exceeds that of $^{13}$C and $^{43}$Ca. Moreover, it is an I = ½ nucleus, leading much more rapidly to quantitative data if relaxation delays are carefully set. It follows that $^1$H is a target nucleus in the study of crystalline hydrated calcium oxalate phases and KS. Moreover, as KS are bio-nanocomposites, $^1$H can be considered as a spectroscopic
spy present both in the organic and inorganic components making the study of the interfaces eventually possible. In the absence of local dynamics, the strong $^1$H–$^1$H dipolar interaction is a major issue in $^1$H solid state NMR leading to considerable broadening of the resonances. Current trends to reach the highest $^1$H NMR resolution combine ultra-fast MAS, up to 111 kHz or above (Samoson, 2019) with ultra-high magnetic field, up to 35.2 T, (Gan et al., 2017) in order to average the strong dipolar couplings. Indeed, the homogeneous character of the homonuclear dipolar interaction implies poor MAS efficiency at low to
moderate spinning frequencies (Schmidt-Rohr and Spiess, 1994) (note that the temperature increase, inside a 0.7 mm diameter rotor, is estimated to roughly 20 °C in the fast/ultra-fast regime, *i.e.* ν$_{rot}$ > 30 kHz. This point is of prime importance as calcium oxalate structures may undergo subtle structural modifications upon heating (Deganello, 1981; Shepelenko, 2019) - see also





section 7). However, such leading-edge equipment is not widely available. An alternative is to use the CRAMPS sequence at moderate spinning frequency ($\nu_{rot}$ < 12 kHz) (Paruzzo and Emsley, 2019). The DUMBO sequence (Decoupling Using Mind-

Boggling Optimization) belongs to the CRAMPS family (Lesage et al., 2003). Using this approach, the internal temperature increase remains moderate for all rotor diameters. Moreover, this methodology can be successfully implemented on almost all magnets. Moreover, larger rotor diameters may be used which can be interesting in terms of sensitivity. To the best of our knowledge, synthetic COM, COD and COT samples were never investigated by [1]H high resolution solid state NMR. The corresponding spectra are presented in Figure 1. At $\nu_{rot}$ = 12 kHz, standard [1]H MAS NMR spectra (Figure 1a) are all

characterized by very broad and almost featureless lineshapes. Such spectral fingerprints are not useful for analytical purposes due to strong overlap of the resonances. DUMBO decoupling leads to a drastic increase in resolution and to very characteristic features for each synthetic hydrate.

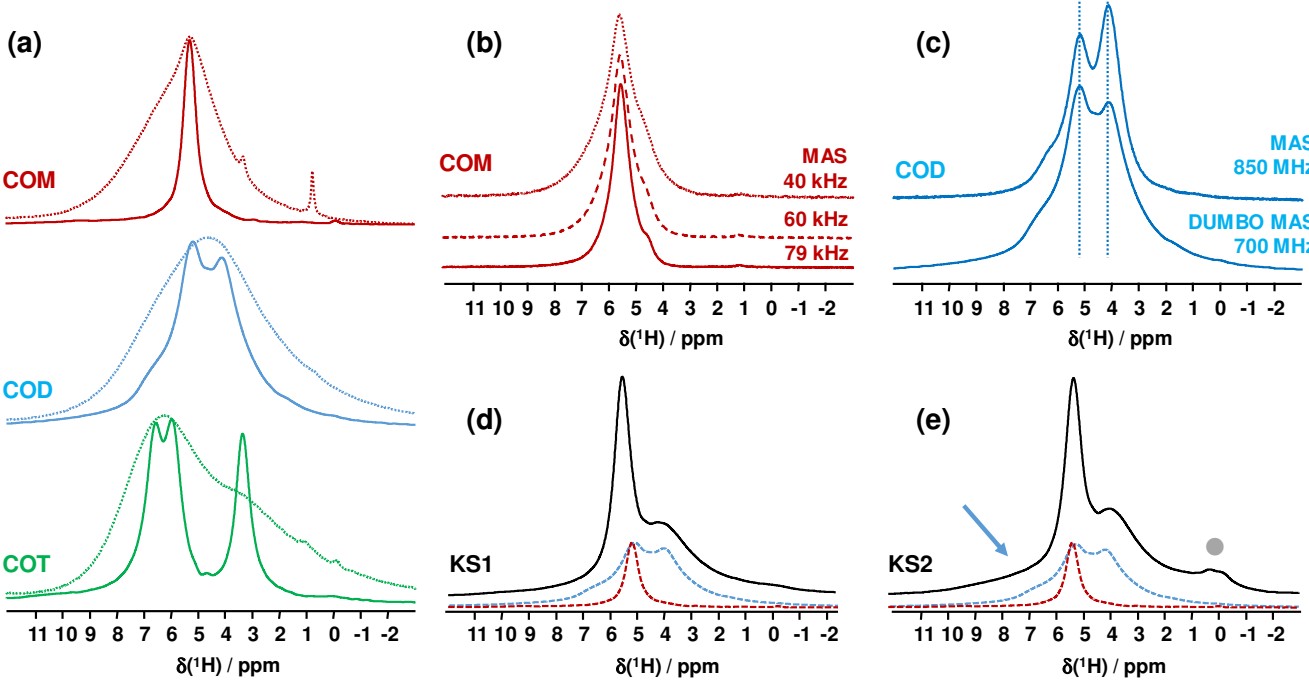

**Figure 1.** (a) [1]H MAS (dashed lines) and [1]H DUMBO MAS (solid lines) NMR spectra of COM (in red), COD (in blue), COT

(in green) ($\nu_{rot}$ = 12 kHz, 700 MHz, 16.4 T). Only the isotropic resonances are represented. (b) [1]H very-fast MAS (40, 60 and 79 kHz) NMR spectra of COM at very high magnetic field (850 MHz). (c) Comparison of the [1]H NMR spectra of COD obtained under DUMBO MAS ($\nu_{rot}$ = 12 kHz, 700 MHz) and very-fast MAS ($\nu_{rot}$ = 79 kHz, 850 MHz) conditions. Vertical dashed lines are guidelines for the eyes. (d) [1]H DUMBO MAS NMR spectrum of KS1 ($\nu_{rot}$ = 12 kHz, 700 MHz). The red and

blue dashed lines correspond to experimental [1]H DUMBO MAS NMR spectra of COM and COD, respectively. (e) [1]H DUMBO MAS NMR spectrum of KS2 ($\nu_{rot}$ = 12 kHz, 700 MHz). The plain light grey circle indicates the presence of organic



components in KS2. The blue arrow indicates the superposition of organic components (aromatic region) and the deshielded shoulder of COD.

The COT crystallographic structure exhibits six inequivalent sites for protons (Heijnen, 1985) whereas only three resonances are clearly observed at $\delta_{iso}(^1H)$ = 3.36, 5.95 and 6.53 ppm (Figure 1a). A realistic assumption is that some resonances are so close that they cannot be distinguished even under DUMBO decoupling. It has been previously shown (Eckert et al., 1988; Pourpoint et al., 2007) that $\delta_{iso}(^1H)$ can be related to the *shortest* O-$\underline{H}$···$\underline{O}$ bond length in hydrogen bond networks. The general trend is that $\delta_{iso}(^1H)$ strongly increases with the shortening of O-$\underline{H}$…$\underline{O}$. Interestingly, the six non equivalent hydrogens can be

distinguished based on O-$\underline{H}$···$\underline{O}$ distances, leading to three distinct groups (Figure 2 and Table S1): 1.668-1.679 Å / 1.809-1.837Å / 1.957-1.978 Å. It is stressed here that the distances were obtained after extensive optimization of the geometry of the COT structure at DFT level (the same comment holds for COM and COD structures, see section 7). According to the literature (Pourpoint et al., 2007), a variation of O-$\underline{H}$···$\underline{O}$ of ~ 0.3 Å is related to a $\delta_{iso.}(^1H)$ variation of ~ 3.5 ppm, in rather good agreement with the results presented here (the shorter the distance, the higher the isotropic $^1H$ chemical shift). We mention

also (Table S1) that each proton of the structure is involved in a relatively high number of $\underline{H}$···$\underline{O}$ contacts (from 3 to 4 with O-$\underline{H}$···$\underline{O}$ ≤ 3 Å). The 3Å cut-off is realistic when considering "weak" H-bonds (Steiner, 2002). In other words, the shortest O-$\underline{H}$···$\underline{O}$ distance dictates directly $\delta_{iso.}(^1H)$, whereas the number of $\underline{H}$···$\underline{O}$ contacts is more representative of the electrostatic/dispersion contributions at a given H position (Steiner, 2002). As all protons of COT are characterized by a large number of $\underline{H}$···$\underline{O}$ contacts, we assume a certain character of "rigidity" of the structure at room temperature and very limited local dynamics (Figure 3c). Under this simple assumption, at most three resolved $\delta_{iso.}(^1H)$ are expected due to similarities in

O-$\underline{H}$···$\underline{O}$ distances (see above), in good agreement with the experimental data (Figure 1a). Therefore, the $^1H$ COT assignments are the following, using the numbering given in Table S1: H1/H6: 3.36 ppm, H3/H5: 5.95 ppm, H2/H4: 6.53 ppm. We note that partial deuteration could be of great help to increase further the resolution.





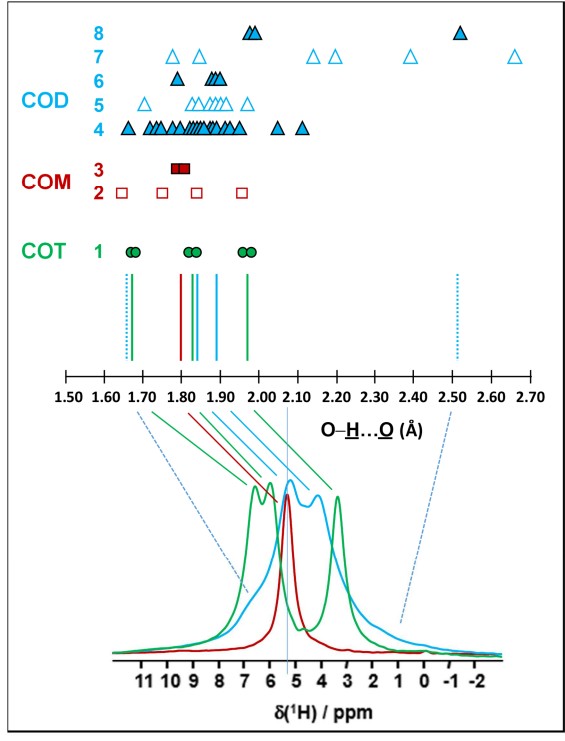

**Figure 2.** Prediction of the relative positions of $\delta_{iso}(^1H)$ for COM (red), COD (blue) and COT (green) as function of the shortest O-$\underline{H}$…$\underline{O}$ distances (in Å). General rules: (i) for a given O-$\underline{H}$…$\underline{O}$ distance, a $\delta_{iso}(^1H)$ is associated (vertical colored solid lines), (ii) if local dynamics are present, averaged O-$\underline{H}$…$\underline{O}$ distances are first calculated. All distances are derived from optimized geometries at the DFT level (Table S1 and section 7). The effect of eventual local dynamics in the case of "less rigid" structure is taken into account. Line1: the structure of COT is considered as "rigid" (plain green circles). On the basis of the shortest O-$\underline{H}$…$\underline{O}$ distance, the six inequivalent protons can be associated in three groups. To each group, a single average $\delta_{iso}(^1H)$ is assigned. A total of three lines for COT is predicted (represented by the three vertical green solid lines). Line 2: the structure of COM is considered as "less rigid" (open squares). Line 3: the corresponding averaged distances are represented by plain squares. A single average $\delta_{iso}(^1H)$ is associated as the averaged distances are very close. A total of one line for COM is predicted. Lines 4 to 8: The COD case: COD exhibits both "rigid" (plain triangles, Line 4) and "less rigid" water molecules (open triangles, Line 5 for the four structural water molecules and Line 7 for the three *zeolitic* water molecules.). Line 6: the corresponding averaged distances for the structural water molecules are represented by plain triangles. Line 8: the corresponding averaged distances for the *zeolitic* water molecules are represented by plain triangles. A continuum of $\delta_{iso}(^1H)$ is predicted for COM. The vertical blue dashed lines correspond to the expected limits of $\delta_{iso}(^1H)$. The two vertical solid blue lines correspond to local maxima adding Lines 4, 6 and 8. Bottom: the superposition of the $^1H$ DUMBO MAS NMR spectra for COM (red), COD (blue) and COT (green) (Figure 1a). The solid and dashed lines connect the experimental data and the predicted $\delta_{iso}(^1H)$.


In the case of COM, the four H crystallographic sites (H11/H12/H21/H22) (Deganello, 1981) are characterized by a large

160 range of O-$\underline{H}$…$\underline{O}$ distances (from 1.647 to 1. 957 Å) and a restricted number of $\underline{H}$···$\underline{O}$ contacts, from 1 to 3 (Table S1). Therefore, a "less rigid" structure is expected at room temperature (when compared to COT). Rapid flips of $H_2O$ molecules could lead to partial averaging of $\delta_{iso}(^1H)$ of protons belonging to the same molecule (Figure 3): using the numbering given in Table S1, the average O-$\underline{H}$···$\underline{O}$ distances for H11/H12 and H21/H22 are very similar (*i.e.* 1.802 and 1.795 Å, respectively): a unique resonance is therefore expected, in full agreement with the $^1H$ DUMBO MAS NMR spectrum of COM (one resonance

165 centered at 5.26 ppm) (Figures 1a and 2). Data obtained at 100K (see Figure S2) demonstrated the presence of four resolved $^1H$ resonances for COM. It is worth noting that $\delta_{iso.}(^1H)$ for H11/H12/H21/H22 in COM and H3/H5 in COT are very close experimentally in agreement with the associated O-$\underline{H}$···$\underline{O}$ distances. From one synthetic sample to the other, $\delta_{iso}(^1H)$ may slightly vary under DUMBO conditions (~ 0.3 ppm). Such a variation is attributed to some disorder of the water molecules which can be present in the COM crystallographic structure (Shepelenko, 2019).

170



**Figure 3.** Structural details of COM (a), COD (b) and COT (c). For each proton of the water molecules, the shortest O-H…O distance (in Å) is represented as well as the number of H…O contacts (cut-off: 3 Å). For COT, the number of H…O contacts is high (3 to 4): the COT structure is considered as "rigid". In the case of COD, the structural and *zeolitic* water molecules are represented in blue and green, respectively. A selection of "rigid" (H4) and "less rigid" (H33/H34) water molecules is presented. All distances and number of H…O contacts are summarized in Table S1. Color code: red: O, black: C, light pink: H.

The $^1$H spectrum of COD (Figure 1a) is *a priori* complex as it corresponds to the superposition of structural and *zeolitic* water molecules (Tazzoli and Domeneghetti, 1980; Izatulina et al., 2014). It is much broader than the spectra corresponding to COM and COT. More specific features centred at $\delta_{iso}(^1H)$ = 4.11, 5.17 and ~ 6.5 (shoulder) ppm are observed (Figure 1a). When compared to the COT spectrum, the spectral resolution decreases as expected from the partial disorder of the *zeolitic* water molecules. The detailed examination of selected O-H···O distances in detail (Table S1) allowed proposing a partial assignment of the resonances. For that purpose, a model (relaxed at the DFT level) which corresponds to $CaC_2O_4$·(2+*0.375*)$H_2O$ was first calculated (or, $Ca_8C_{16}O_{32}(H_2O)_{16}(H_2O)_3$). The water molecules located in the channels of the *zeolitic* structure are represented in *italic*. Taking into account the number of H···O contacts (in full analogy with the approach described above for COM and COT), among the 19 water molecules, 7 molecules are considered as "less rigid" (or, potentially mobile) out of which 4 of them are structural and *3* are *zeolitic*. The remaining 12 water molecules are considered as "rigid". Typical example of "rigid" (H4) and "less rigid" (H33/H34) water molecules are presented in Figure 3b. From Figure 2, it is then possible to predict the expected ranges of $\delta_{iso}(^1H)$ for COD. The rules applied are that "rigid" water molecules correspond to two distinct $\delta_{iso}(^1H)$ (Line 4), whereas "less rigid" water molecules correspond to a single average $\delta_{iso}(^1H)$ (averaging of Line 5 gives Line 6 and averaging of Line7 gives Line 8). The sum of Lines 4, 6 and 8 (in blue) corresponds to the expected $^1$H spectrum for COD. On this basis, it is expected: (i) that $\delta_{iso}(^1H)$ is distributed over a much larger range when compared to COM and COT, (ii) that some maxima should be observed (at least two, corresponding to large number of overlapping triangles in Figure 2). Points (i) and (ii) are in very good agreement with experimental observations. All in all, the relative predicted positions for $\delta_{iso}(^1H)$ resonances are in agreement with the experimental data for COM, COD and COT (bottom of Figure 2), validating the proposed assignments.

As a first conclusion of this section, $^1$H DUMBO MAS NMR spectra for COM, COD and COT correspond to useful fingerprints for analytical purposes as they are clearly characteristic for each phase. Such fingerprints can be used for the analysis of $^1$H NMR spectra of KS (see below). We emphasize that $^1$H spectra with excellent signal-to-noise ratio were obtained within minutes. As $\delta_{iso.}(^1H)$ values are very sensitive to H-bond networks as well as to local motional averaging, studies performed on synthetic COM, COD and COT were necessary prior to the detailed analyses of KS. Nevertheless, some comments have to be made at this stage: (i) First, the $^1$H DUMBO MAS methodology ($\nu_{rot}$ = 12 kHz, 700 MHz) is comparable to the very-fast MAS/very-high magnetic field approach ($\nu_{rot}$ ~ 80 kHz, 850 MHz) without any multiple pulses decoupling. In



Figure 1b, the $^1$H MAS NMR spectra of COM are presented at various fast/very-fast rotation frequencies, from $\nu_{rot}$ = 40 to
~ 80 kHz. As in the case of the DUMBO MAS approach, a single resonance (with a small shoulder) was observed, showing a
continuously decreasing linewidth with increasing the MAS frequency. The linewidth obtained at ~ 80 kHz is still broader
than the one observed under DUMBO MAS conditions. In Figure 1c, the two approaches are compared in the case of COD.
The resolution is slightly enhanced under very-fast MAS at 79 kHz but remains comparable to DUMBO conditions at 12 kHz.
More importantly, the relative intensities are not strictly preserved indicating that some distortions of the lineshapes may occur
under DUMBO conditions. It follows that only semi-quantitative data can be extracted at best in the case of complex mixtures
of hydrated calcium oxalate phases. Moreover, dynamics at room temperature may impact the efficiency of the DUMBO
decoupling.

Finally, two KS (KS1 and KS2) were studied by $^1$H DUMBO MAS NMR (Figures 1d and 1e). In the case of KS1, a mixture
of COM and COD is immediately detected (in full agreement with FTIR and powder XRD, not shown here). As stated above,
a slight deviation of $\delta_{iso}(^1H)$ for COM is observed. A semi-quantitative analysis of the COM/COD proportions is possible and
could be systematically compared to FITR analyses (as routinely obtained in hospitals). In addition to COM and COD
resonances, the $^1$H DUMBO MAS NMR spectrum of KS2 exhibits small new contributions that can be attributed to organic
moieties (such as proteins). In this case, a semi-quantitative analysis of the KS appears more difficult to perform. Working at
much higher magnetic field, 35.2 T (Gan et al., 2017), and under ultra-fast MAS (>> 100 kHz) should lead to increased
resolution and easier direct quantification of the spectra.

## 2.2 $T_2*(^1H)$ editing and $^1H–^1H$ DQ experiments

Another KS sample, KS3, that was studied contained a large proportion of organic moieties (as shown by FTIR). Another
option to increase $^1$H NMR resolution is to implement standard Hahn echoes with increasing delays, $\tau$, (up to several ms) and
synchronization with the MAS frequency. The magnetization associated to protons characterized by short $T_2*(^1H)$ will dephase
very rapidly. $^1$H MAS echoes ($\nu_{rot}$ = 30 kHz) for KS3 are presented in Figure 4. From powder XRD data (not shown here)
confirmed by $^{13}$C CP MAS NMR data (see section 4), KS3 contains COM as a major mineral phase. For long $\tau$, sharp lines
(associated to mobile components) were obtained whereas the broader COM component around 5.2 ppm was totally supressed.
$\delta_{iso}(^1H)$ values agree with unsaturated fatty acids ($\delta_{iso}(^1H)$ ~ 5.25 ppm) (Ren et al., 2008). The presence of triglycerides is
excluded as the C$\underline{H}$ and C$\underline{H}_2$ resonances of the glycerol backbone ($\delta_{iso}(^1H)$ ~ 5.0 and 4.0 ppm, respectively) were not detected.





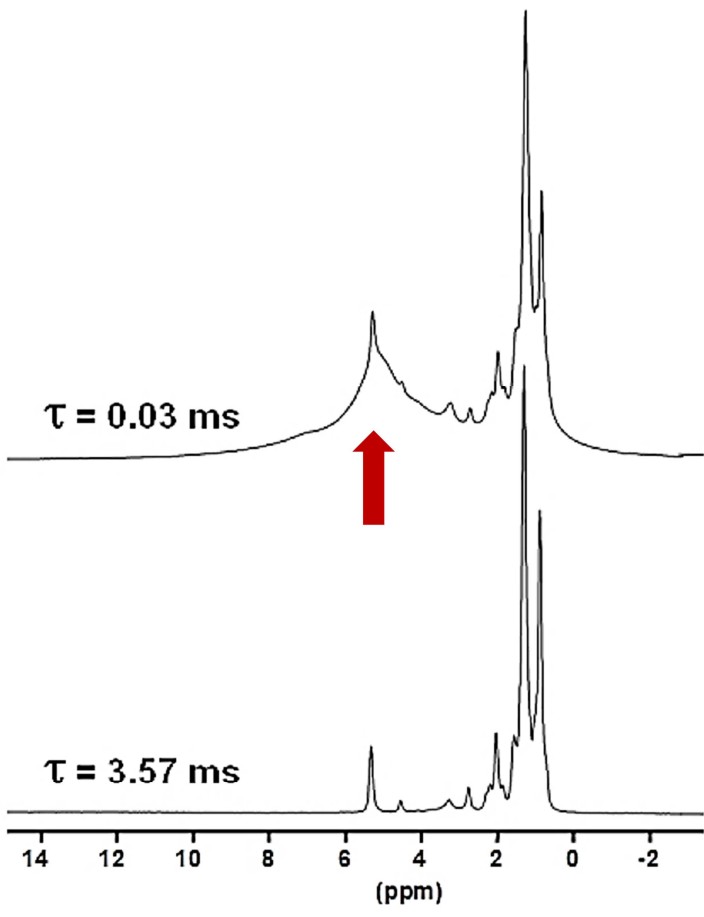

**Figure 4.** $^1$H Hahn echo MAS NMR spectra for KS3 recorded at 16.4 T. $\tau$ was synchronized with the rotation frequency, here $\nu_{rot}$ = 30 kHz. No temperature control was implemented leading to a ~ 40 °C increase of the sample temperature and the associated increase of local dynamics. The vertical red arrow corresponds to the resonance coming from COM (see also Figures 1a and 1b).

Such level of resolution allowed for the implementation of J-MAS derived pulse schemes such as the $^1$H−$^1$H DQF COSY MAS experiment (based on isotropic J($^1$H−$^1$H) couplings). This experiment is part of the toolbox for more general dynamics-based spectral editing research topic, applied to biological solids (Mroue et al., 2016; Matlahov and van der Wel, 2018; Gopinath and Veglia, 2018). The $^1$H−$^1$H DQF COSY MAS spectrum is presented in Figure 5a for KS3. All resonances of the mobile fatty acid chains were assigned in a straightforward way demonstrating the pertinence of this *through bond* correlation

experiment. On the other hand, dipolar based double quantum (DQ) experiments can be implemented to establish *through space* proximities between protons (such as BABA, Back to Back) (Feike et al., 1996). It is a distinct advantage to perform such experiments under very-fast MAS (here 79 kHz). Indeed, the spectral resolution is drastically increased leading to an





easier observation of the correlation peaks. The $^1$H-$^1$H DQ BABA MAS NMR spectrum of KS3 is presented in Figure 5b. The $^1$H resonance corresponding to the COM phase is clearly evidenced on the $^1$H projection and on the 2D diagonal (red arrows).

Moreover, red dashed ovals indicate correlations involving the protons of the immobile proteins contained in KS3 (essentially the $^1$H$^N$–$^1$H$^\alpha$, $^1$H$^\alpha$–$^1$H$^\beta$ regions). $^1$H spin diffusion experiments should help to highlight actual correlations between the organic and inorganic components, at the interface (Schmidt-Rohr and Spiess, 1994).

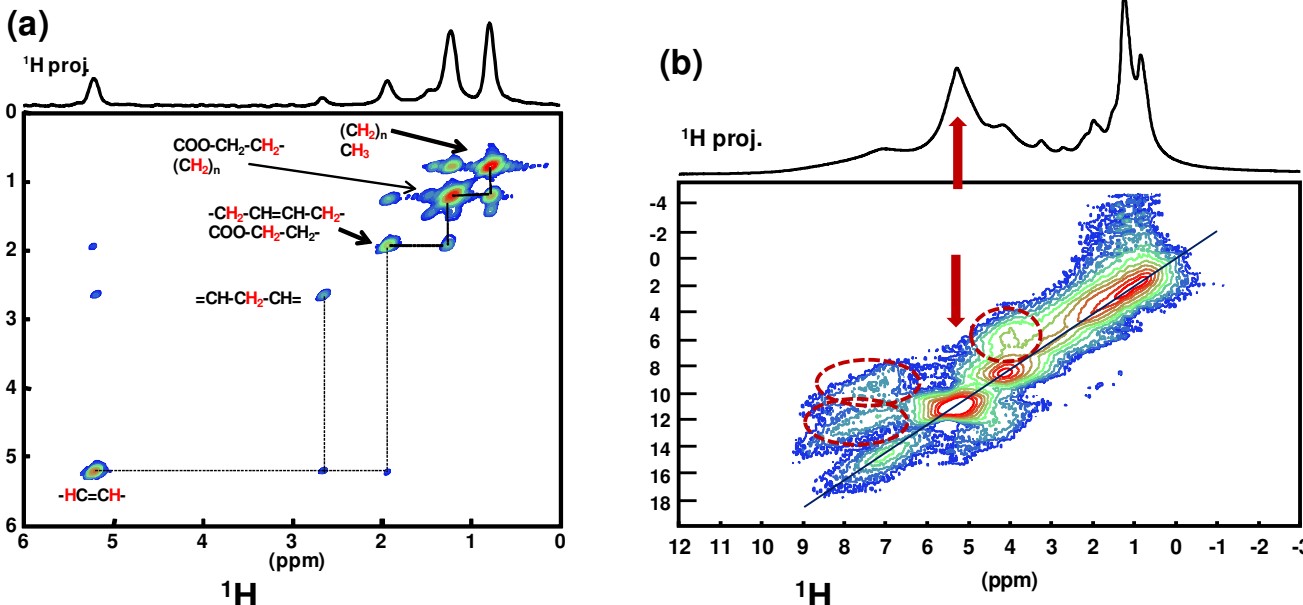

**Figure 5.** (a) $^1$H-$^1$H DQF COSY MAS NMR spectrum for KS3 at $\nu_{rot}$ = 30 kHz recorded at 16.4 T. Here, no temperature control was implemented leading to an ~ 40 °C increase of the local temperature and therefore of local dynamics. All peaks are assigned to contributions from unsaturated mobile fatty acids (with unsaturations). (b) $^1$H-$^1$H DQ BABA MAS NMR spectrum for KS3 at $\nu_{rot}$ = 79 kHz recorded at 16.4 T (no temperature control). Recoupling period: 2 rotor periods. Off-diagonal correlations (immobile organic moieties) are highlighted by dashed red ovals. The red arrows indicate the COM contribution.

**3 Natural abundance $^{43}$Ca solid state NMR experiments**

*Natural abundance* solid state $^{43}$Ca MAS NMR spectroscopy remains a challenge. Indeed, the NMR characteristics of this quadrupolar nucleus (I = 7/2) are clearly unfavourable: natural abundance: 0.14 % and low $\gamma$ ($\nu_0$ = 57.2 MHz at 20 T). Nevertheless four main experimental approaches have been successfully developed during the last few years: (i) using large volume rotors (7 mm, ~ 400 mg of sample) at high magnetic field (20 T), under moderate MAS (~ 5 kHz) and implementing

DFS (Double Frequency Sweep) excitation scheme (section 7), (ii) using much smaller rotors (3.2 mm, ~ 20 mg of sample at





ultra-high magnetic field (35.2 T) and under moderate/fast MAS (~ 18 kHz), (iii) using Dynamic Nuclear Polarization (DNP) to strongly enhance the $^{43}$Ca polarization, usually in the indirect mode (from $^1$H to $^{43}$Ca), (iv) labeling in $^{43}$Ca (starting from an enriched calcite precursor) (Laurencin et al., 2021; Smith, 2020; Laurencin and Smith, 2013). Here, we follow the approach (i) which is by far the easiest to implement in most NMR facilities worldwide (as long as a low-γ probe is available).

The first contributions related to the study of synthetic calcium oxalates hydrates by $^{43}$Ca MAS NMR spectroscopy were proposed by Wong for COT (Wong et al., 2006) and by Bowers and Kirkpatrick for the three hydrated phases (Bowers and Kirkpatrick, 2011). The latter claimed that the COM lineshape could be attributed to an averaged Gaussian signal due to a local disorder in the structure (Tazzoli and Domeneghetti, 1980). Colas (Colas et al., 2013) demonstrated that high signal-to- noise ratio is necessary to extract reliable quadrupolar parameters from natural abundance $^{43}$Ca MAS NMR spectra

and re-investigated the COM phase. Instead of a Gaussian contribution, two distinct resonances were evidenced, in agreement with the crystallographic data ($\delta_{iso}(^{43}$Ca$) = -2.6$ ppm, $C_Q = 1.50$ MHz, $\eta_Q = 0.60$; $\delta_{iso}(^{43}$Ca$) = 0.7$ ppm, $C_Q = 1.60$ MHz, $\eta_Q = 0.70$). The $^{43}$Ca MAS NMR spectra of COM, COD and COT recorded at 20.0 T are presented in Figure 6a. All spectra were obtained in natural abundance in a reasonable amount of experimental time (~ 2 hours for COM and COT, ~ 4 hours for COD). The $^{43}$Ca NMR fingerprints obtained allows unambiguous distinctions of the three phases. The sharpest line

(characterized by the smallest $C_Q$) is observed for COT (one unique crystallographic site). For this particular phase, second-order quadrupolar broadening is efficiently suppressed at 20 T, leading directly to $\delta_{iso}(^{43}$Ca$) = -0.1$ ppm. This value is slightly different from the one reported by Wong (*i.e.* $-4.2$ ppm) (Wong et al., 2006). Such a discrepancy can be attributed to a difference in chemical shift referencing (Gervais et al., 2008). The associated quadrupolar parameters for COT (Wong et al., 2006) were $C_Q = 1.55$ MHz, $\eta_Q = 0.72$. $C_Q$ is probably overestimated as such value would definitely produce second-order

quadrupolar broadening under MAS at 20.0 T (see above the quadrupolar parameters for COM). Finally, a rather featureless spectrum is obtained for COD (one crystallographic site), exhibiting a much larger linewidth than for COT ($\delta_{iso}(^{43}$Ca$) \sim -2.6$ ppm, $C_Q \sim 1.60$ MHz, $\eta_Q \sim 0.20$). We assign this broadening to the distribution of *zeolitic* water molecules (leading consequently to a slight distribution of $\delta_{iso}(^{43}$Ca$)$). Hence, it is demonstrated that natural abundance $^{43}$Ca MAS NMR spectroscopy is useful in characterizing hydrated calcium oxalate phases. The use of moderate MAS is sufficient to retrieve

satisfactory resolution as characteristic $C_Q(^{43}$Ca$)$ are usually small/very small (< 1.8 MHz). However, the $\delta_{iso}(^{43}$Ca$)$ range covered by these three phases is small making this NMR parameter less sensitive distinguishing the hydrates. This comes from the fact that $\delta_{iso}(^{43}$Ca$)$ is mainly determined by the coordination of the Ca atoms and the mean <Ca-O> distances. These parameters are almost identical for COM, COD and COT (8-fold coordination for COM and COD, 7-fold coordination for COT, range of averaged Ca–O distances: 2.47–2.49 Å. This last comment is rather in contradiction with previous conclusions

proposed in the literature (Bowers and Kirkpatrick, 2011).





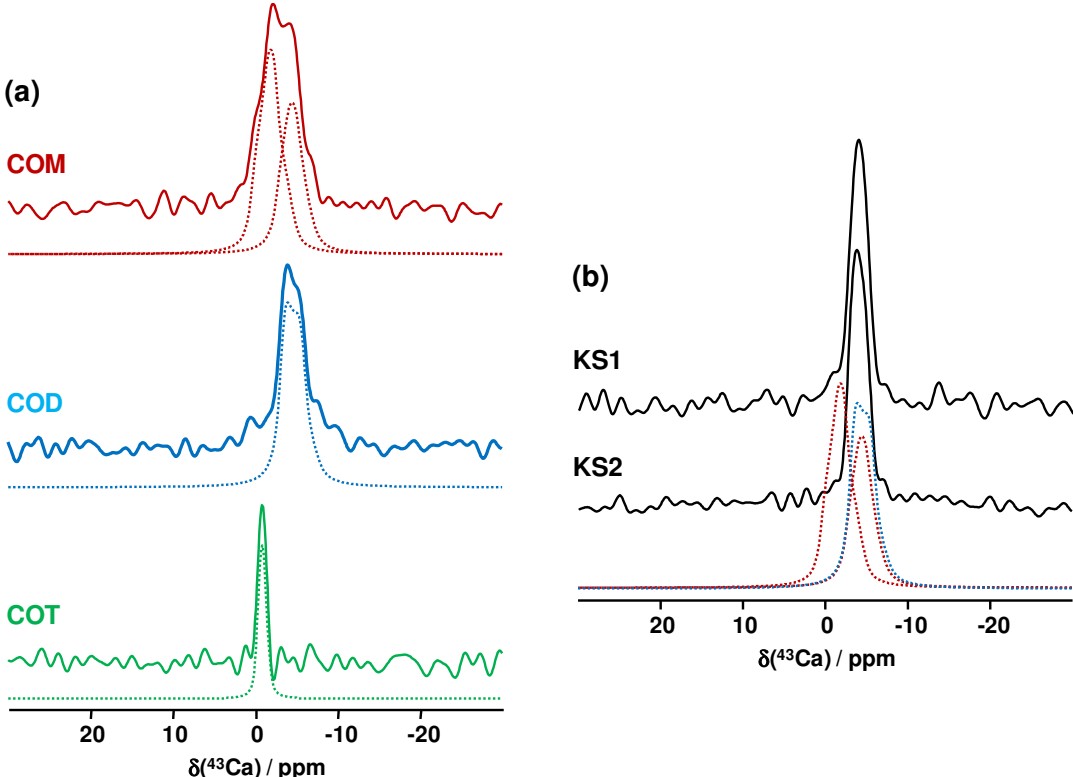

**Figure 6.** (a) Natural abundance $^{43}$Ca MAS NMR spectra of COM (red), COD (blue) and COT (green) recorded at 20.0 T ($\nu_{rot}$ = 3 to 5 kHz). The dashed lines correspond to fits. (b) Natural abundance $^{43}$Ca MAS NMR spectra of KS1 and KS2. The
red dashed lines correspond to the two resonances associated to COM. The blue dashed line corresponds to the $^{43}$Ca MAS NMR spectrum of COD.

The natural abundance $^{43}$Ca MAS NMR spectra of KS1 and KS2 are presented in Figure 6b. They are largely similar to the COD spectrum overall. The contribution of a COM component is hardly discernable (though present, especially in KS1, see
Figure 7). As stated above, the structure of COM is subject to subtle structural variations which could lead to the overlap of the two $^{43}$Ca resonances. In other words, though interesting in principle, natural abundance $^{43}$Ca MAS NMR spectroscopy (associated inherently to limited signal-to-noise ratio) should not be used as a first solid state NMR tool of investigation for KS.

## 4 Back to $^{13}$C NMR: spectral edition and reconstruction of spectra

$^{13}$C NMR data related to synthetic calcium oxalate phases and KS are the most represented in the literature. This is probably due to the fact that the spectral resolution is high under MAS and that CP (cross polarization) MAS experiments can easily be



implemented even at low or moderate magnetic field. Typical [13]C CP MAS NMR spectra for COM and COD are presented in Figure 7 (bottom). Four isotropic resonances are observed for COM as expected from XRD data (Colas et al., 2013) and one unique broader resonance is observed for COD as expected from XRD data considering the disorder associated to the *zeolitic* water molecules. Such disorder has an impact on the resolution of the [13]C NMR spectra.

RESONANCE
Discussions

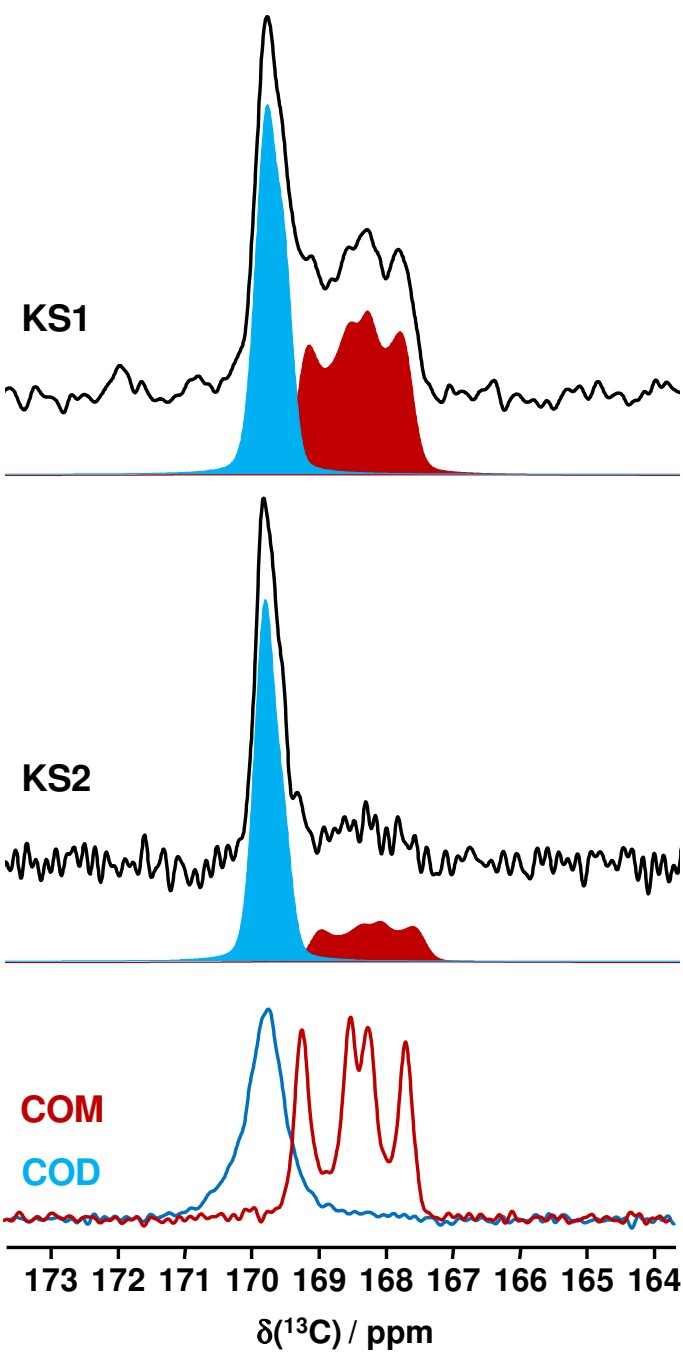

**Figure 7.** $^{13}$C CP MAS NMR spectra of KS1 and KS2 (recorded at 16.4 T, $\nu_{rot}$ = 5 kHz) and the corresponding COM (red) and COD (blue) contributions. Bottom: $^{13}$C CP MAS NMR spectra of synthetic COM (red line) and COD (blue line) recorded under similar conditions.




It is observed that the chemical shift range of interest is very restricted (~ 4 ppm from 167 to 171 ppm) corresponding to ~ 0.8% of the whole $^{13}$C isotropic chemical shift range. $^{13}$C CP MAS NMR spectra for KS1 and KS2 are also presented in Figure 7. The presence of COM and COD components is clearly evidenced and could be quantified if necessary (by increasing

the signal-to-noise ratio significantly). As a matter of fact, a single experiment at fixed contact time (usually > 5 ms) is sufficient in principle for quantitative purposes as $^1$H–$^{13}$C dipolar couplings are comparable for all $^{13}$C sites (differences in relative intensities can be evidenced at much short contact time, *i.e.* < 0.5 ms). The case of KS3 is by far more complex. As stated in section 1, a given KS may include a complex organic component, containing lipids, triglycerides, membrane components, glycoproteins (like the Tamm-Horsfall protein) and glycoaminoglycans, among other species (Reid et al., 2011).

The approximate chemical composition of KS3 is: ~ 10 % proteins, ~ 20-25 % COM and ~ 65 % amorphous silica (Dessombz et al., 2016). In Figure 8, we propose a robust protocol to reconstruct the $^{13}$C MAS NMR spectra starting from well identified sub-spectra. At short contact time (0.8 ms), all carbon-containing species are detected, corresponding to both sharp and broad lines (Figure 8a).

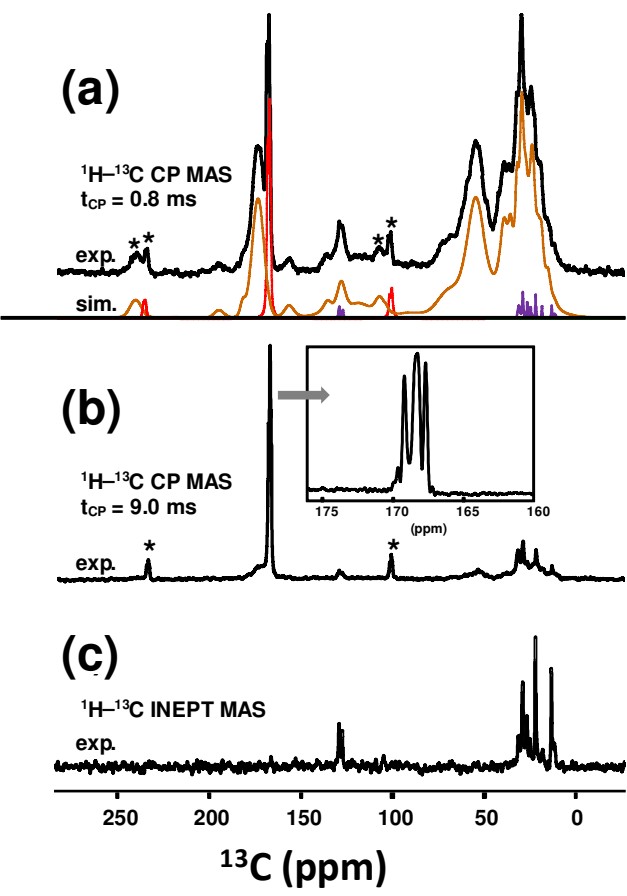




**Figure 8.** (a) $^{13}$C CP MAS NMR spectrum of KS3 (recorded at 7.0 T using a short contact time, 0.8 ms, $\nu_{rot}$ = 5 kHz). The experimental spectrum is decomposed in three components: COM (in red), fatty acids (in purple) and proteins (in brown). (b) $^{13}$C CP MAS NMR spectrum of KS3 (recorded at 7.0 T using a long contact time, 9.0 ms). The insert highlights the COM contribution (four resonances, two of them being almost overlapped) (Colas et al. 2013). (c) $^{1}$H-$^{13}$C refocused INEPT J-MAS NMR spectrum of KS3 (recorded at 7.0 T). The unsaturations of the fatty acids are clearly evidenced at $\delta_{iso}(^{13}C)$ ~ 130 ppm. *: spinning sidebands.

Then, a $T_{1\rho}(^{1}H)$ filter was applied by increasing the contact time by a factor of ~ 10 leading to the drastic reduction of the intensities of the broad components. The four resonances of COM are clearly observed (insert in Figure 8b). COD is absent in agreement with powder XRD and FTIR data. It follows that the proton spin baths corresponding to COM and the broad components are independent (spin diffusion and domain size measurements could be implemented as complementary experiments (Schmidt-Rohr and Spiess, 1994). The 1D $^{1}$H-$^{13}$C refocused INEPT J-MAS NMR sequence (Figure 8c) allowed selective extraction of the mobile components corresponding to the fatty acids (see also Figure 5a). The unsaturated nature is clearly evidenced by the shift at $\delta_{iso}(^{13}C)$ ~ 130 ppm. Finally, the $^{13}$C CP MAS NMR spectrum (Figure 8a, bottom) could be reconstructed with resonances from: (i) the COM phase and its associated spinning sidebands (in red), (ii) fatty acids characterized by very sharp lines (in purple), (iii) and proteins (in brown) (Cavanagh et al., 2007) for which a precise attribution cannot be given at this stage.

## 5 The ubiquitous (but elusive) presence of phosphorus in KS: $^{31}$P MAS and CP MAS experiments

Bak *et al.* (Bak et al., 2000) used $^{31}$P MAS and CP MAS experiments to evidence phosphate-containing phases in KS. The presence of phosphate groups in KS is not unusual and observed mainly by FTIR (Figure S1). However, their exact chemical nature remains unclear. Phosphates in KS can correspond to: (i) mineral phases such as substituted (carbonated) hydroxyapatite ($Ca_{10}(PO_4)_6(OH)_2$), brushite ($CaHPO_4 \cdot 2H_2O$) or struvite ($NH_4MgPO_4 \cdot 6H_2O$), (ii) organic phosphates present in phospholipids (in the cell membrane) and/or DNA, RNA, ATP molecules (Butusov and Jernelöv, 2013). Usually, phosphates are found as minor components in KS making $^{31}$P NMR attractive given the high inherent signal sensitivity of $^{31}$P (which is also an I = ½ nucleus). A total of six KS (exhibiting COM as the major phase and the "apparent" absence of phosphate phases by powder XRD) were studied here. The representative $^{31}$P MAS and CP MAS NMR spectra of the KS are presented in Figure 9.





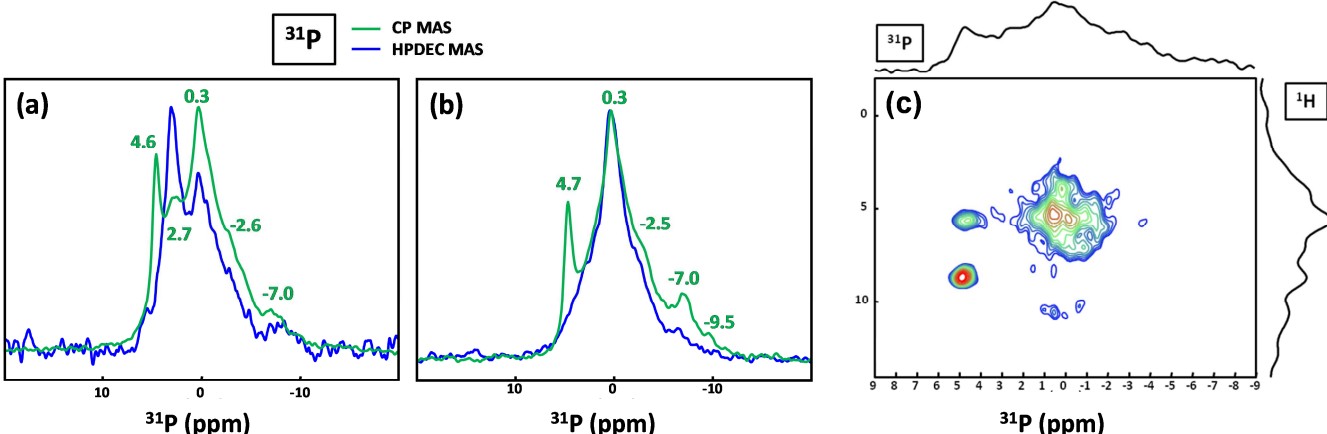

**Figure 9.** (a) $^{31}$P MAS under high power {$^1$H} decoupling (in blue) and CP MAS (in green) NMR spectra of KS4. Some specific chemical shifts are highlighted. (b) $^{31}$P MAS under high power {$^1$H} decoupling (in blue) and CP MAS (in green) NMR spectra of KS5 (representative of an ensemble of five KS). Some specific chemical shifts are highlighted. (c) $^1$H–$^{31}$P HETCOR CP MAS NMR spectrum of KS4 (temperature control at −20 °C). All spectra shown here were recorded at 16.4 T.

The $^{31}$P NMR fingerprint of KS4 is specific (Figure 9a), whereas KS5 has a $^{31}$P fingerprint analogous to four other KS (Figure 9b). The acquisition time is ~ 2 to 3 hours demonstrating that the amount of phosphate species is indeed small in all samples. One notes a large distribution of $\delta_{iso}(^{31}$P), corresponding not only to structural disorder, but also to strong chemical variability. In order to facilitate the assignment of $\delta_{iso}(^{31}$P), $^1$H–$^{31}$P HETCOR CP MAS NMR experiments under active temperature control (T = −20 °C) were implemented as well (Figure 9c). Three clear correlations were observed: $\delta_{iso}(^{31}$P) = 4.6 ppm ↔ $\delta_{iso}(^1$H) = 8.7 ppm; $\delta_{iso}(^{31}$P) = 4.6 ppm ↔ $\delta_{iso}(^1$H) = 5.7 ppm, $\delta_{iso}(^{31}$P) ~ 0.25-0.30 ppm ↔ $\delta_{iso}(^1$H) = ~ 5.0 ppm. Reasonable assignments are the following (Godinot et al., 2016): (i) The peak centered at $\delta_{iso}(^{31}$P) = 4.6 ppm is assigned to struvite, NH$_4$MgPO$_4$·6H$_2$O (Bak et al., 2000). The correlation centred at $\delta_{iso}(^{31}$P) = 4.6 ppm ↔ $\delta_{iso}(^1$H) = 8.7 ppm (ammonium groups) is attributed to PO$_4^{3-}$/NH$_4^+$. The correlation centred at $\delta_{iso}(^{31}$P) = 4.6 ppm ↔ $\delta_{iso}(^1$H) = 5.7 ppm concerns water molecules. It is interesting to note that the amount of struvite is extremely small (almost absent in the $^{31}$P MAS NMR spectrum of KS4 and KS5 - Figures 9a and 9b). (ii) The resonance at $\delta_{iso}(^{31}$P) = 0.3 ppm may be attributed to phosphates in phospholipids (in this case $\delta_{iso}(^{31}$P) is in the ~ 1 to −1 ppm range). However, correlations with $\delta_{iso}(^1$H) < 3 ppm are almost absent (such resonances should be characteristic for long alkyl chains in phospholipids). Consequently, we assign the $^{31}$P resonance to inorganic (hydrated) orthophosphates. (iii) $\delta_{iso}(^{31}$P) ~ 2.7 ppm could be potentially assigned to amorphous calcium phosphate with a rather small (rather unusual) level of protonation (this resonance is underestimated in the CP MAS experiment, Figure 9a). (iv) $\delta_{iso}(^{31}$P) << 0 ppm resonances are assigned to pyro- and/or polyphosphates.



## 6 Conclusions and perspectives

This study has demonstrated that the solid state NMR technique offers a complementary characterisation approach for the study of kidney stones and related synthetic model systems. The $^1$H DUMBO MAS NMR technique provides unambiguous identification of the different calcium oxalate hydrate phases. This experiment is a rapid-measurement technique which can be easily adapted to yield semi-quantitative data. For the first time, the natural abundance $^{43}$Ca MAS NMR data from the three calcium oxalate hydrate phases have been presented together; these data exhibited sufficient signal-to-noise to facilitate a complete structural interpretation in agreement with crystallographic data. The extension of this approach to the study of KS was attempted showing that a real signal could be measured, but with relatively limited discrimination between the different KS samples. The deconvolution of the $^1$H and $^{13}$C MAS NMR data into assigned sub-spectra aided the interpretation of the data describing the whole system, thus demonstrating that KS materials are usually a complex association of organic and inorganic components. Additional $^{31}$P MAS NMR studies provided further insight into the composition of the low-level phosphates which are ubiquitous and difficult to characterize in KS. The development of solid state NMR, in combination with modern computational DFT and Machine Learning approaches, would be able to characterize the complex heterogeneous biomaterials such as KS without ambiguity (Tielens et al., 2021). As part of on-going studies building on the observations here, systematic NMR studies of a large range of KS from the Tenon Hospital's collection is being undertaken to develop new diagnosis NMR approaches that could impact on developing novel treatments.

## 7 Syntheses of hydrated calcium oxalate, kidney stones samples and NMR methods

*Synthesis*. Calcium chloride (CaCl$_2$) and sodium oxalate (Na$_2$C$_2$O$_4$) were purchased from Sigma-Aldrich and used as received. All syntheses were carried out using distilled water. COM: at 40°C, equimolar aqueous solutions of Na$_2$C$_2$O$_4$ and CaCl$_2$ (0.1 mol.L$^{-1}$) were added simultaneously dropwise in a few mL of water under magnetic stirring. The mixture was left mixing under these conditions during 2 hours before filtration and was then washed with cold water before drying under air. COD: a Na$_2$C$_2$O$_4$ aqueous solution (0.1 mol.L$^{-1}$) and a CaCl$_2$ solution (1.0 mol.L$^{-1}$, Ca/Ox = 10) were prepared the day prior to the reaction and stored between 2–6 °C overnight. The solution of Na$_2$C$_2$O$_4$ was added dropwise to the CaCl$_2$ solution in an ice bath (T < 7°C) under magnetic stirring. The mixture was left under stirring for 15 min before filtration and was then washed with cold water before drying under air. COT: in an ice bath, two equimolar (0.001 mol.L$^{-1}$) aqueous solutions of Na$_2$C$_2$O$_4$ and CaCl$_2$ were slowly added simultaneously dropwise in a few mL of water under vigorous magnetic stirring. The mixture was left under stirring for 15 min before filtration and was then washed with cold water before drying under air. All COM, COD, COT samples were obtained as white fine powders. COD and COT were rapidly stored between 2–6 °C while COM could be stored at ambient temperature. *Kidney stones*. The samples were provided by Dr M Daudon (Tenon Hospital, Paris, France). The choice of the diameter of the used NMR rotor was dictated by the initial size of the KS and the implemented





experiments. In the case of large KS, smaller pieces were studied as powders by NMR. *NMR methods. Warning*: the COM structure is highly sensitive to temperature variations ($\geq 15$ °C). The lowest MAS frequencies have to be implemented for all investigated nuclei as well as active regulation of the sample temperature (Bruker BCUX unit). Most of the $^1$H MAS and DUMBO MAS NMR spectra presented in Figure 1 were obtained at 700 MHz (Bruker AVANCE III spectrometer) using a

2.5 mm Bruker MAS probe spinning the sample at 12 kHz (number of scans: 20 to 40, recycle delay: 10 s for quantitative measurements, temperature: 10 °C, $t_{90°}(^1H) = 3.0$ µs, duration of the shape length: 24 µs at 113 kHz RF field). The DUMBO experiment was first set up with glycine as a test sample (including the scaling of the isotropic chemical shift) and then optimized for each compound. Some $^1$H MAS NMR spectra were obtained at 850 MHz using a 1mm JEOL MAS probe (spinning the sample up to 79 kHz) (number of scans: 4, recycle delay: 3 s, $t90°(^1H) = 1.70$ µs). Synchronized Hahn echoes

(Figure 4) were performed at 700 MHz using a 2.5 mm Bruker MAS probe spinning the sample at 30 kHz (number of scans: 64, recycle delay: 5 s, $t_{90°}(^1H) = 2.8$ µs, no active regulation of the temperature in order to increase local dynamics – the increase in temperature is estimated to ~ 40 °C). The $^1$H–$^1$H DQF COSY MAS NMR experiment (Figure 5) was performed at 700 MHz using a 2.5mm Bruker MAS probe at 30 kHz (number of scans: 32, recycle delay: 2 s, $t_{90°}(^1H) = 2.8$ µs, 256 increments in $t_1$ dimension, no active regulation of the temperature in order to increase local dynamics, magnitude mode). The $^1$H–$^1$H SQ-DQ

BABA MAS NMR experiment (Figure 5) was performed at 850 MHz using a 1 mm JEOL MAS NMR probe spinning the sample at 79 kHz (number of scans: 16, recycle delay: 3 s, $t90°(^1H) = 1.70$ µs, 2 BABA loops, 426 increments in $t_1$ dimension, no active regulation of the temperature). All $^1$H NMR spectra were referenced using adamantane (1.85 ppm) as a secondary reference. All natural abundance $^{43}$Ca NMR spectra (Figure 6) were obtained at 850 MHz (Bruker AVANCE III spectrometer) using a 7 mm low-γ Bruker MAS single channel NMR probe spinning the sample at 3 to 5 kHz. A DFS (Double Frequency

Sweep) (Iuga et al., 2000) enhancement scheme followed by a 90° selective pulse of 1.5 µs, was used (DFS pulse length of 2 ms, RF ~ 8 kHz, and convergence sweep from 400 to 50 kHz, number of scans: from 5600 to 18000, recycle delay: 0.8 s). All $^{43}$Ca chemical shifts were referenced at 0.0 ppm to a 1.0 mol.L$^{-1}$ aqueous solution of $CaCl_2$ (Gervais et al., 2008). The $^1$H–$^{13}$C RAMP CP MAS experiments (Figure 7) were obtained at 700 MHz (Bruker AVANCE III spectrometer) using a 2.5 mm Bruker MAS double resonance NMR probe spinning the sample at 5 kHz (number of scans: 600 to 1200, recycle delay: 3 s,

$t_{90°}(^1H) = 3.1$ µs, contact time: 2 to 8 ms). The $^{13}$C MAS NMR spectra presented in Figure 8 were obtained at 300 MHz (Bruker AVANCE III spectrometer) using a 7 mm Bruker MAS double resonance NMR probe spinning the sample at 5 kHz (number of scans: 328, recycle delay: 3 s, $t90°(^1H) = 5.2$ µs, contact time: 0.8 and 9.0 ms, refocused INEPT MAS: number of scans: 6000, recycle delay: 3 s, 5.2 and 3.2 µs π/2 pulse on $^1$H and $^{13}$C respectively, no active regulation of the temperature). All $^{13}$C NMR spectra were referenced using adamantane (38.48 ppm) as a secondary reference. $^{31}$P 1D and 2D NMR spectra

presented in Figure 9 were obtained at 700 MHz (Bruker AVANCE III spectrometer) using a 2.5 mm Bruker MAS double resonance NMR probe spinning the sample at 30 kHz (number of scans: ≈ 4000 for high power {$^1$H} decoupling experiments and ≈ 3700 for CP MAS experiments, recycle delay: 10 s for high power {$^1$H} decoupling experiments and flip angle: 30°, 3 s for CP MAS experiments, $t_{90°}(^1H) = 2.0$ µs, contact time for CP MAS experiments: 5.0 ms). For the $^1$H–$^{31}$P HETCOR RAMP



CP MAS experiment: number of scans: 400, recycle delay: 3 s, $t_{90°}(^1H)$ = 2.0 µs, contact time: 5.0 ms, 96 increments in $t_1$
dimension, active regulation of the temperature at −20 °C.

*Relaxation of crystallographic structures.* Starting from the crystallographic data, COM (Daudon et al., 2009), COD (Tazzoli and Domeneghetti, 1980) and COT (Basso et al., 1997) structures were relaxed at DFT level. The unit cell parameters as well as the atomic positions were optimized as previously described for COM (Colas et al., 2013). VASP was used (Kresse and Hafner, 1993; Kresse and Hafner, 1994; Kresse and Furthmüller, 1996). The corresponding CIF files are available upon
request.

## 8 Appendices

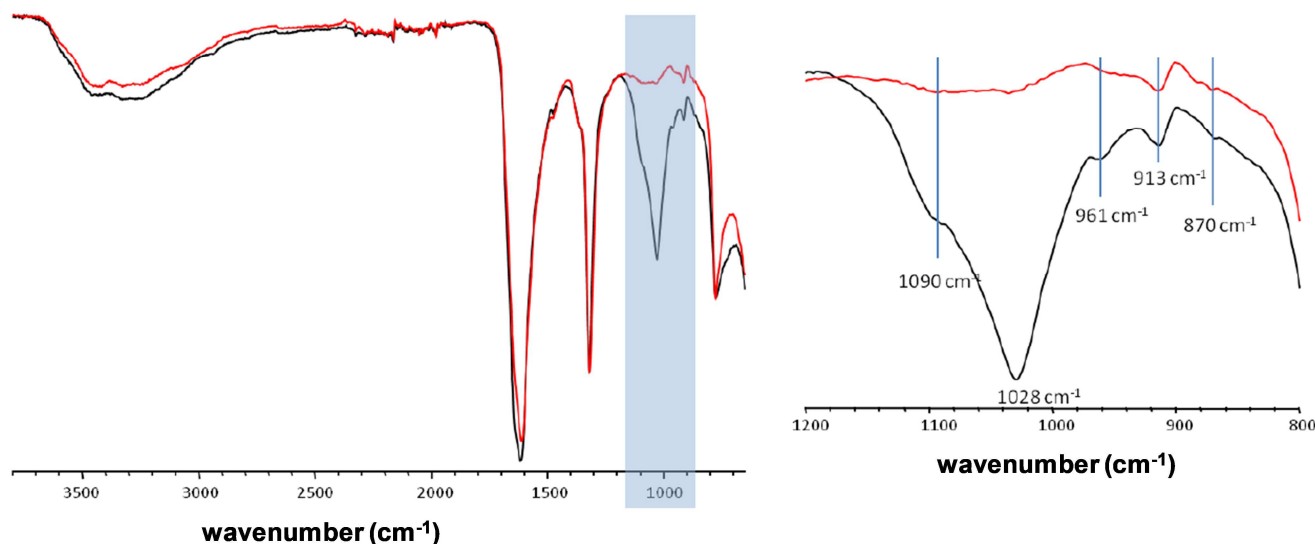

**Figure A1**. FTIR spectra of two KS containing a mixture of COM and COD phases. The main difference lies in the light blue
wavenumber region corresponding to phosphate vibrations (including hydroxyapatite: 913, 961, 1090 cm$^{-1}$, carbonates: 870 cm$^{-1}$).



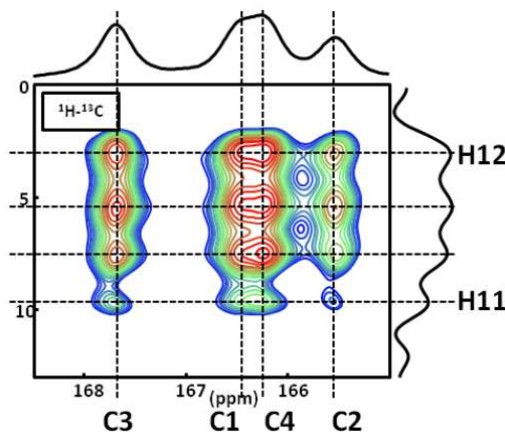

**Figure A2**. $^1H-^{13}C$ CP DNP MAS NMR spectrum of COM at T = 100 K. Four distinct $^1H$ resonances are clearly evidenced on the $^1H$ indirect dimension. The contact time is 9.0 ms and sixteen $^1H/^{13}C$ correlations are observed.


**Table A1**. All CIF files are available upon request for COM, COD and COT.

(a) selected O-H…O distances and number of H…O contacts with O-H…O ≤ 3Å (highlighted in yellow) for COM. The atomic positions were optimized at the DFT level.

| H11 | O100 | 1.00849 | H12 | O100 | 0.98369 | H21 | O200 | 0.99077 | H22 | O200 | 0.98971 |
|-----|------|---------|-----|------|---------|-----|------|---------|-----|------|---------|
| H11 | H12 | 1.57350 | H12 | H11 | 1.57350 | H21 | H22 | 1.60332 | H22 | H21 | 1.60332 |
| H11 | O8 | 1.64743 | H12 | O5 | 1.95727 | H21 | O5 | 1.75421 | H22 | O100 | 1.84259 |
| H11 | C4 | 2.32231 | H12 | H22 | 2.34911 | H21 | C3 | 2.41467 | H22 | H12 | 2.34911 |
| H11 | H22 | 2.38986 | H12 | C3 | 2.43432 | H21 | O6 | 2.59359 | H22 | H11 | 2.38986 |
| H11 | O7 | 2.54004 | H12 | O1 | 2.82575 | H21 | Ca1 | 2.91456 | | | |
| H11 | Ca2 | 2.97148 | H12 | Ca2 | 2.90392 | H21 | H12 | 2.92563 | | | |
| | | | H12 | H21 | 2.92563 | | | | | | |
| | | | H12 | O8 | 2.97218 | | | | | | |

(b) selected O-H…O distances and number of H…O contacts with O-H…O ≤ 3Å (highlighted in yellow) for COD. The atomic positions were optimized at the DFT level. In order to take into account the distribution of the *zeolitic* water molecules, a model corresponding to $CaC_2O_4.(2+0.375)H_2O$ was first calculated (see below) - $Ca_8C_{16}O_{32}(H_2O)_{16}(H_2O)_3$. The water molecules located in the channels of the *zeolitic* structure are represented in *italics*. The "less rigid" molecules are highlighted in red rectangles. The first 4 molecules are structural. The last 3 molecules are *zeolitic*.



**MAGNETIC RESONANCE** — Open Access — Discussions

| | | | | | | | | | | | | | | | | | | | | |
|---|---|---|---|---|---|---|---|---|---|---|---|---|---|---|---|---|---|---|---|---|
| H1 | O33 | 0.98706 | H2 | O34 | 0.98980 | H3 | O34 | 0.98182 | H4 | O33 | 0.99442 | H5 | O35 | 0.98170 | H6 | O36 | 0.98695 | H7 | O36 | 0.98579 |
| H1 | H4 | 1.59565 | H2 | H3 | 1.57549 | H3 | H2 | 1.57549 | H4 | H1 | 1.59565 | H5 | H8 | 1.61227 | H6 | H7 | 1.58205 | H7 | H6 | 1.58205 |
| H1 | O26 | 1.91539 | H2 | O25 | 1.82678 | H3 | O28 | 2.04985 | H4 | O27 | 1.75128 | H5 | O30 | 1.85900 | H6 | O29 | 1.87561 | H7 | O32 | 1.91845 |
| H1 | O17 | 2.11120 | H2 | H18 | 2.12287 | H3 | H19 | 2.22917 | H4 | H38 | 1.97874 | H5 | O14 | 2.75907 | H6 | H22 | 2.15494 | H7 | H23 | 2.15003 |
| H1 | H38 | 2.35789 | H2 | H37 | 2.24223 | H3 | H37 | 2.66372 | H4 | H20 | 2.04373 | H5 | H21 | 2.15627 | H6 | H37 | 2.31981 | H7 | H22 | 2.69686 |
| H1 | H20 | 2.69799 | H2 | H19 | 2.67477 | H3 | H18 | 2.70041 | H4 | H17 | 2.52979 | H5 | C10 | 2.81281 | H6 | H23 | 2.64503 | H7 | O16 | 2.78683 |
| H1 | O15 | 2.84946 | H2 | H8 | 2.74763 | H3 | O12 | 2.81150 | H4 | O44 | 2.63337 | H5 | H24 | 2.83299 | H6 | O42 | 2.77692 | H7 | O42 | 2.79486 |
| H1 | O10 | 2.79429 | H2 | O9 | 2.75952 | H3 | O43 | 2.84290 | H4 | C13 | 2.68644 | H5 | O41 | 2.88697 | H6 | O13 | 2.79229 | H7 | H37 | 2.81901 |
| H1 | C16 | 2.86535 | H2 | C15 | 2.77043 | H3 | H8 | 2.86354 | H4 | O11 | 2.84953 | H5 | Ca3 | 2.93317 | H6 | C9 | 2.81462 | H7 | C12 | 2.85808 |
| H1 | Ca1 | 2.99269 | H2 | O43 | 2.77691 | H3 | C14 | 2.95471 | H4 | O51 | 2.90084 | | | | H6 | Ca4 | 2.92805 | H7 | Ca4 | 2.95463 |
| | | | H2 | O51 | 2.78975 | H3 | Ca2 | 2.98530 | H4 | O4 | 2.93927 | | | | H6 | O6 | 2.99753 | | | |
| | | | H2 | Ca2 | 2.93267 | | | | H4 | Ca1 | 2.96741 | | | | | | | | | |

| | | | | | | | | | | | | | | | | | | | | |
|---|---|---|---|---|---|---|---|---|---|---|---|---|---|---|---|---|---|---|---|---|
| H8 | O35 | 1.00238 | H9 | O37 | 0.98455 | H10 | O38 | 0.98386 | H11 | O38 | 0.98764 | H12 | O37 | 0.98525 | H13 | O39 | 0.98603 | H14 | O40 | 0.98212 |
| H8 | H5 | 1.61227 | H9 | H12 | 1.57848 | H10 | H11 | 1.59461 | H11 | H10 | 1.59461 | H12 | H9 | 1.57848 | H13 | H16 | 1.60309 | H14 | H15 | 1.57384 |
| H8 | O51 | 1.66198 | H9 | O18 | 1.89267 | H10 | O17 | 1.85718 | H11 | O50 | 1.85667 | H12 | O19 | 1.84639 | H13 | O50 | 1.83852 | H14 | O21 | 2.11357 |
| H8 | H38 | 1.99179 | H9 | H25 | 2.14808 | H10 | H26 | 2.12211 | H11 | H36 | 2.17741 | H12 | H28 | 2.07478 | H13 | H36 | 2.35234 | H14 | H30 | 2.23119 |
| H8 | H37 | 2.04976 | H9 | H36 | 2.36458 | H10 | H33 | 2.34664 | H11 | H13 | 2.40635 | H12 | H34 | 2.50717 | H13 | H11 | 2.40635 | H14 | H36 | 2.65940 |
| H8 | H2 | 2.74763 | H9 | H28 | 2.64453 | H10 | H27 | 2.65917 | H11 | H35 | 2.60426 | H12 | H25 | 2.62744 | H13 | H35 | 2.56253 | H14 | H31 | 2.70447 |
| H8 | H3 | 2.86354 | H9 | O48 | 2.77037 | H10 | O47 | 2.76488 | H11 | O20 | 2.76299 | H12 | O48 | 2.71565 | H13 | O22 | 2.81506 | H14 | O5 | 2.79807 |
| H8 | H24 | 2.93412 | H9 | O2 | 2.78591 | H10 | C7 | 2.80358 | H11 | O39 | 2.81858 | H12 | O3 | 2.76892 | H13 | H29 | 2.85749 | H14 | O46 | 2.83725 |
| H8 | O31 | 2.96498 | H9 | C8 | 2.84076 | H10 | O1 | 2.88960 | H11 | O4 | 2.82251 | H12 | C5 | 2.83943 | H13 | O6 | 2.93190 | H14 | O38 | 2.89969 |
| H8 | O34 | 2.99827 | H9 | Ca5 | 2.93707 | H10 | O39 | 2.92353 | H11 | H27 | 2.85030 | H12 | Ca5 | 2.95153 | | | | H14 | H33 | 2.94927 |
| | | | H9 | O50 | 2.98631 | H10 | Ca6 | 2.98407 | | | | H12 | H36 | 2.99920 | | | | | | |

| | | | | | | | | | | | | | | | | | | | | |
|---|---|---|---|---|---|---|---|---|---|---|---|---|---|---|---|---|---|---|---|---|
| H15 | O40 | 0.99513 | H16 | O39 | 0.98321 | H17 | O44 | 0.98418 | H18 | O43 | 0.98306 | H19 | O43 | 0.98845 | H20 | O44 | 0.98141 | H21 | O41 | 0.98524 |
| H15 | H14 | 1.57384 | H16 | H13 | 1.60309 | H17 | H20 | 1.58638 | H18 | H19 | 1.58312 | H19 | H18 | 1.58312 | H20 | H17 | 1.58638 | H21 | H24 | 1.59032 |
| H15 | O24 | 1.77827 | H16 | O23 | 1.89139 | H17 | O26 | 1.90254 | H18 | H2 | 2.12287 | H19 | O28 | 1.83208 | H20 | O27 | 1.97101 | H21 | O30 | 1.84549 |
| H15 | H31 | 2.12464 | H16 | H32 | 2.16066 | H17 | H1 | 2.11120 | H18 | C15 | 2.52659 | H19 | H3 | 2.22917 | H20 | H4 | 2.04373 | H21 | H5 | 2.15627 |
| H15 | H36 | 2.19988 | H16 | H29 | 2.69091 | H17 | C16 | 2.50781 | H18 | H3 | 2.70041 | H19 | C14 | 2.48695 | H20 | C13 | 2.60434 | H21 | C10 | 2.45463 |
| H15 | H30 | 2.67989 | H16 | C3 | 2.79378 | H17 | H4 | 2.53979 | H18 | O34 | 2.78745 | H19 | H2 | 2.67477 | H20 | H1 | 2.69799 | H21 | O35 | 2.74039 |
| H15 | C4 | 2.70528 | H16 | O45 | 2.80838 | H17 | O33 | 2.71797 | | | | H19 | O34 | 2.82331 | H20 | O33 | 2.77138 | | | |
| H15 | O8 | 2.75027 | H16 | O7 | 2.82588 | H17 | O1 | 2.96592 | | | | | | | | | | | | |
| H15 | O46 | 2.77041 | H16 | H33 | 2.95661 | | | | | | | | | | | | | | | |
| H15 | O50 | 2.91170 | H16 | Ca7 | 2.98301 | | | | | | | | | | | | | | | |

| | | | | | | | | | | | | | | | | | | | | |
|---|---|---|---|---|---|---|---|---|---|---|---|---|---|---|---|---|---|---|---|---|
| H22 | O42 | 0.98588 | H23 | O42 | 0.98660 | H24 | O41 | 0.99754 | H25 | O48 | 0.98498 | H26 | O47 | 0.98243 | H27 | O47 | 0.99450 | H28 | O48 | 0.98088 |
| H22 | H23 | 1.58163 | H23 | H22 | 1.58163 | H24 | H21 | 1.59032 | H25 | H28 | 1.57886 | H26 | H27 | 1.58899 | H27 | H26 | 1.58899 | H28 | H25 | 1.57886 |
| H22 | O29 | 1.91449 | H23 | O32 | 1.87699 | H24 | O31 | 1.70417 | H25 | O18 | 1.89272 | H26 | O17 | 1.88627 | H27 | O20 | 1.73992 | H28 | O19 | 1.95113 |
| H22 | H6 | 2.15494 | H23 | H7 | 2.15003 | H24 | C11 | 2.43596 | H25 | H9 | 2.14808 | H26 | H10 | 2.12211 | H27 | C6 | 2.43921 | H28 | H12 | 2.07478 |
| H22 | C9 | 2.54313 | H23 | C12 | 2.50035 | H24 | H5 | 2.83299 | H25 | C8 | 2.53078 | H26 | C7 | 2.49549 | H27 | H10 | 2.65917 | H28 | C5 | 2.52406 |
| H22 | H7 | 2.69686 | H23 | H6 | 2.64503 | H24 | O35 | 2.93124 | H25 | H12 | 2.62744 | H26 | O7 | 2.69101 | H27 | O6 | 2.68011 | H28 | H9 | 2.64453 |
| H22 | O36 | 2.82367 | H23 | O36 | 2.79218 | H24 | H8 | 2.93412 | H25 | O8 | 2.73239 | H26 | Ca7 | 2.86542 | H27 | Ca7 | 2.83851 | H28 | O5 | 2.71256 |
| | | | H23 | O7 | 2.97205 | | | | H25 | O37 | 2.78268 | H26 | O38 | 2.88787 | H27 | H11 | 2.85030 | H28 | O37 | 2.74612 |
| | | | | | | | | | H25 | Ca8 | 2.87171 | | | | H27 | O38 | 2.98955 | H28 | Ca8 | 2.91374 |

| | | | | | | | | | | | |
|---|---|---|---|---|---|---|---|---|---|---|---|
| H29 | O45 | 0.99748 | H30 | O46 | 0.99081 | H31 | O46 | 0.98265 | H32 | O45 | 0.98460 |
| H29 | H32 | 1.58925 | H30 | H31 | 1.58606 | H31 | H30 | 1.58606 | H32 | H29 | 1.58925 |
| H29 | O22 | 1.72144 | H30 | O21 | 1.80278 | H31 | O24 | 1.92821 | H32 | O23 | 1.85294 |
| H29 | C2 | 2.42196 | H30 | H14 | 2.23119 | H31 | H15 | 2.12464 | H32 | H16 | 2.16066 |
| H29 | O2 | 2.65935 | H30 | C1 | 2.47815 | H31 | C4 | 2.55193 | H32 | C3 | 2.49151 |
| H29 | H16 | 2.69091 | H30 | H15 | 2.67389 | H31 | H14 | 2.70447 | H32 | O3 | 2.71237 |
| H29 | Ca5 | 2.83343 | H30 | O1 | 2.71669 | H31 | O4 | 2.74962 | H32 | O39 | 2.81984 |
| H29 | H13 | 2.85749 | H30 | O40 | 2.85346 | H31 | O40 | 2.82082 | H32 | Ca5 | 2.85738 |
| H29 | O39 | 2.87558 | H30 | Ca6 | 2.85690 | H31 | Ca6 | 2.88812 | | | |
| | | | H30 | O14 | 2.87981 | | | | | | |


| | | | | | | | | | | | |
|---|---|---|---|---|---|---|---|---|---|---|---|
| H33 | O49 | 0.97686 | H34 | O49 | 0.97560 | H35 | O50 | 0.98605 | H36 | O50 | 0.97905 |
| H33 | H34 | 1.55468 | H34 | H33 | 1.55468 | H35 | H36 | 1.58568 | H36 | H35 | 1.58568 |
| H33 | H10 | 2.34664 | H34 | O19 | 2.39218 | H35 | O49 | 1.84598 | H36 | O40 | 2.14165 |
| H33 | H35 | 2.58196 | H34 | H12 | 2.50717 | H35 | H34 | 2.53067 | H36 | H11 | 2.17741 |
| H33 | O38 | 2.66257 | H34 | H35 | 2.53067 | H35 | H13 | 2.56253 | H36 | H15 | 2.19988 |
| H33 | O17 | 2.72932 | H34 | C5 | 2.65374 | H35 | H33 | 2.58196 | H36 | H13 | 2.35234 |
| H33 | H14 | 2.94927 | H34 | C3 | 2.91240 | H35 | H11 | 2.60426 | H36 | H9 | 2.36458 |
| H33 | H16 | 2.95661 | H34 | O3 | 2.96977 | | | | H36 | O37 | 2.61085 |
| H33 | C7 | 2.98511 | | | | | | | H36 | H14 | 2.65940 |
| | | | | | | | | | H36 | O38 | 2.89925 |
| | | | | | | | | | H36 | H12 | 2.99920 |

| | | | | | |
|---|---|---|---|---|---|
| H37 | O51 | 0.97973 | H38 | O51 | 0.99205 |
| H37 | H38 | 1.55005 | H38 | H37 | 1.55005 |
| H37 | H8 | 2.04976 | H38 | O33 | 1.77829 |
| H37 | O34 | 2.20072 | H38 | H4 | 1.97874 |
| H37 | H2 | 2.24223 | H38 | H8 | 1.99179 |
| H37 | H6 | 2.31981 | H38 | H1 | 2.35789 |
| H37 | O36 | 2.45695 | | | |
| H37 | H3 | 2.66372 | | | |
| H37 | H7 | 2.81901 | | | |

Ca$_8$C$_{16}$O$_{32}$(H$_2$O)$_{16}$(*H$_2$O)$_3$*: structural water molecules are represented in blue, *zeolitic* molecules are represented in green.

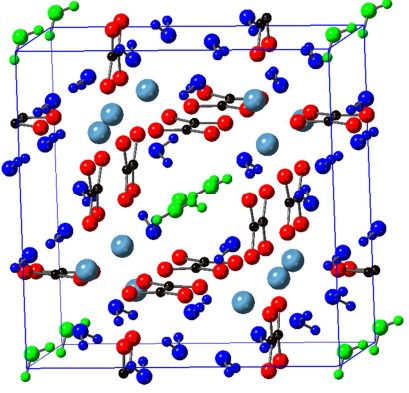






(c) selected O-H…O distances and number of H…O contacts with O-H…O ≤ 3Å (highlighted in yellow) for COT. The atomic positions were optimized at the DFT level.

| H1 | O6 | 0.98194 | H2 | O6 | 1.01013 | H3 | O7 | 0.98972 | H4 | O5 | 1.00726 | H5 | O5 | 0.99237 | H6 | O7 | 0.98705 |
|---|---|---|---|---|---|---|---|---|---|---|---|---|---|---|---|---|---|
| H1 | H2 | 1.62553 | H2 | H1 | 1.62553 | H3 | H6 | 1.57438 | H4 | H5 | 1.61032 | H5 | H4 | 1.61032 | H6 | H3 | 1.57438 |
| H1 | O3 | 1.97834 | H2 | O1 | 1.66830 | H3 | O6 | 1.83746 | H4 | O2 | 1.67978 | H5 | O3 | 1.81868 | H6 | O5 | 1.95753 |
| H1 | H3 | 2.09660 | H2 | H3 | 2.15532 | H3 | H1 | 2.09660 | H4 | H6 | 2.08633 | H5 | H1 | 2.37604 | H6 | H4 | 2.08633 |
| H1 | H5 | 2.37604 | H2 | C1 | 2.61465 | H3 | H2 | 2.15532 | H4 | C1 | 2.60506 | H5 | C2 | 2.54822 | H6 | H5 | 2.66412 |
| H1 | C2 | 2.73894 | H2 | H4 | 2.75756 | H3 | H1 | 2.82312 | H4 | H2 | 2.75756 | H5 | H6 | 2.66412 | H6 | O2 | 2.80700 |
| H1 | H3 | 2.82312 | H2 | O4 | 2.79380 | H3 | O3 | 2.87563 | H4 | O7 | 2.83920 | H5 | O4 | 2.77186 | H6 | H4 | 2.84626 |
| H1 | O7 | 2.84845 | H2 | O4 | 2.96096 | H3 | O6 | 2.96615 | H4 | H6 | 2.84626 | H5 | O2 | 2.84989 | H6 | O2 | 2.85413 |
| H1 | O7 | 2.91450 | | | | H3 | Ca1 | 2.98521 | H4 | O7 | 2.85001 | H5 | Ca1 | 2.91831 | | | |
| H1 | H4 | 2.93242 | | | | | | | H4 | H1 | 2.93242 | | | | | | |
| H1 | O5 | 2.93253 | | | | | | | H4 | O1 | 2.99648 | | | | | | |
| H1 | H1 | 2.95802 | | | | | | | | | | | | | | | |

**Data Availability.** All the data are shown in the figures of the paper. CIF files of COM, COD, COT structures: available upon request from the corresponding author.

**Author Contributions.** CL performed all syntheses and recorded most of the NMR spectra in strong collaboration with CB, DL and DI. CG and FT performed all DFT optimizations. FB and LB-C were deeply involved in the interpretation of the NMR spectra as well as MES and JVH. MD, EL and DB provided the KS sample and interpreted the data as physicians and a physical 490 chemist, respectively. CB wrote the article in connection with his co-authors.

**Competing Interests.** The authors declare that they have no conflict of interest.

**Acknowledgements.** DFT calculations were performed using HPC resources from GENCI-IDRIS (Grant 097535). The UK 850 MHz solid-state NMR Facility used in this research was funded by EPSRC and BBSRC (contract reference PR140003), as well as the University of Warwick including via part funding through Birmingham Science City Advanced Materials 495 Projects 1 and 2 supported by Advantage West Midlands (AWM) and the European Regional Development Fund (ERDF).

**Financial Support.** CL was funded by an Ecole Doctorale (ED 397) PhD fellowship of Sorbonne University.

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
