# Peer review of "A novel multinuclear solid state NMR approach for the characterization of kidney stones"

_Magnetic Resonance, 2021_

## Author Response (AR1)

We want to thank the anonymous reviewer 1 for his/her helpful comments. *We answer all questions/comments below.*
* * *
- Page 7, line 168. I do not understand the sentence 'from one synthetic sample …… (Shepelenko 2019)'. Does this mean that the COM synthesis is not perfectly reproducible?

*The synthesis is perfectly reproducible in the sense that COM is always obtained by using the described synthetic protocol. No COD or COT were obtained. However, the detailed structure of the obtained COM includes a degree of subtility. It can evolve from an ordered structure (space group P2$_1$/c) mainly described in the literature and a so-called disordered phase exhibiting a statistical I2/m space group (Shepelenko, 2019) (with a priori different NMR characteristics - NB: the powder XRD patterns are almost identical with tiny differences hardly discernable). The relative energies of both phases are comparable meaning that the final COM structure depends strongly on the experimental conditions. The study of the impact of the experimental conditions is out the scope of this contribution. The following sentence has been added in the new version of the MS:*

*"COM is always obtained as a final product as shown by powder XRD. Subtle variations are observed depending on the degree of disorder present as demonstrated very recently by Shepelenko (Shepelenko, 2019).*

- 5b should be represented in a SQ-SQ way to be more easily compared to Fig.5a.

*It is a good idea. It is now included in Figure 5b.*

- 6. Give the experimental details: such as the number of scan and recycling delays.

*All NMR parameters (including NS and RD) are given in section 7 for all experiments/Figures.*

Why not an indirect detection through $^1$H?

*From our experience in natural abundance (0.14%!) $^{43}$Ca MAS NMR spectroscopy, the rather short T$_1$($^{43}$Ca)'s are a clear advantage. It is why direct detection is usually performed.*

In Figs.16b, the correlation of KS1 and KS2 resonances with the COM and COD decompositions is only a hypothesis.

*Indeed. It leads to our conclusion that $^{43}$Ca NMR has not to be used as a first tool of investigation for KS studies. I tis already stated in the MS.*

- 9-caption. The first sentence is not clear. Is the CPMAS spectrum also recorded under high-power $^1$H decoupling?

*Yes, it is. It will be specified in the final version.*

- Part N° 7: sometimes t90°($^1$H), sometimes $t_{90°}$($^1$H).

*It will be carefully checked.*

- Sx or Fig.Ax?

*We will move to the Ax notation.*

- Globally, most figures lack of experimental parameters.

*See above my comment related to section 7.*
* * *
**We want to thank reviewer 2 for his/her helpful comments.** *Our answers are given below*.

**General Comments**:

This manuscript addresses the characterization of kidney stones (KS) – complex calcium oxalate composites of variable structure and composition – focusing on the molecular composition and structure of primarily the biomineral. The central methodology devised herein is multinuclear ssNMR spectroscopy combined with DFT calculations, as essential ingredients with X-ray diffraction and FTIR.

The authors synthesized the three hydrate standards – mono-, di- and tri-hydrate calcium oxalates (COM, COD, COT) and carried out their NMR characterization as a basis to identify the CO (calcium oxalate) content in the KSs.

Initially the authors screen a variety of possible $^1$H MAS NMR techniques and parameters ($^1$H-$^1$H homonuclear decoupling – DUMBO, fast- to very fast-MAS and field strengths); following, they conclude that the DUMBO sequence is best suited to discriminate (resolve) the CO content and apply that one also the two KS1 and KS2 samples in which they identify COM and COD as the major components. A detailed DFT study of the standards is used to correlate the isotropic $^1$H chemical shifts with the hydrogen bond lengths and assign the water peaks to belong to structural, dynamic or zeolitic type of water content in these materials.

Secondly, natural abundance $^{43}$Ca MAS NMR (at 20 Tesla) was applied to the standards and the KS (1 and 2) samples. While the $^{43}$Ca MAS NMR appeared to discriminate between the standards, it showed in the KS samples primarily COD content (contrary to $^1$H and $^{13}$C data). This issue may have to be further studied in the future as to enhance its analytic capability.

*Yes: it will be specified in the final version of the MS:*

*"43Ca NMR would benefit working at ultra-high magnetic field (35T – Ref: Bonhomme et al., Chem. Commun. 2018)in order to drastically increase the resolution and enhance 43Ca NMR analytic capabilities".*

Thirdly, $^{13}$C CP MAS NMR spectra clearly distinguish the COM and COD standards and identifies their occurrence as mixtures of different proportions in the KS1 and KS2 samples. The proportions seem to the naked eye different than seen in the $^{1}$H spectra. A point that may be further discussed in the MS.

*This is a very good comment that will be emphasized in the final version of the manuscript. The following sentence has been added in the final MS:*

*"DUMBO experiments may be sensitive to local dynamics, especially in the intermediate regime (Ref: Emsley et al., J. Magn. Reson. 2019). It may involve some discrepancies between $^{1}$H and $^{13}$C NMR data in terms of quantification".*

*Moreover in the case of KS2, an organic component (proteins,…) is clearly present making the quantification more difficult (it is already mentioned in the main text).*

Information on the organic content present in a third KS sample (KS3) was obtained by $T_2(^{1}H)$ filtering and $^{1}$H-$^{1}$H 2D DQF (all MAS) experiments whose spectra showed a small fraction of (highly mobile) unsaturated fatty acids and bulkier proteinaceous content. This insight was further refined using $T_{1\rho}(^{1}H)$ filtering ($^{13}$C CP MAS) and $^{1}$H-$^{13}$C INEPT which allowed to select the highly mobile components. Finally, representative data on P-content was shown for two other KSs (4 and 5) ascribing it primarily to inorganic hydrated orthophosphates.

I find this work broad and of importance to the biomaterials community, highlighting a glossary of ssNMR techniques and demonstrating analytical capabilities (as well as limitations) to analyze the complex and diverse composites of KSs. Certainly, this MS emphasizes the crucial role of ssNMR as a unique molecular-level complement to the more common and far less detailed techniques.

*Thank you.*

I find this MS suitable for publication after a minor revision.

**Specific Comments**:

As quite a large number of KS samples were examined and as not all were subjected to all characterization techniques, I find it instructive to illustrate the variability of organic content and include a Figure (Appendix) which shows all KS's (for which spectra are available) with full range $^{13}$C CP MAS spectra (250 ppm). It appears that the limited range spectra of the KS samples in Fig. 7 are deficient of organic content – were they measured with 9ms contact time? please note that in caption. In such a case also a comparison of the limited range would be desirable to show extent of robustness of identification for the different KSs.

*This is a very interesting suggestion. A new Figure A3 is added in SI, including the contact time used for each experiment.*

If similar information is available for the P-content throughout the different KSs, it will be as instructive to have it presented as well. From my limited experience with KSs, P-content was not negligible as described herein (line 370) " The acquisition time is ~ 2 to 3 hours demonstrating that the amount of phosphate species is indeed small in all samples ".

*A new Figure A4 is added in SI as well. The experimental time is added in order to get a qualitative idea of the P content for the various KS.*

**Technical Corrections**:

Line 101: Spinning induced temp. increase is referred to in numerous places in the MS (as this may affect the CO); herein I suggest to state how much is moderate? 20C ? 10C ?

*It will be specified for the probes we used.  For 2.5 mm probe: < 5°C at 5 kHz, 40 °C at 30 kHz. For 7 mm probe: order of magnitude: 5 °C at 5 kHz.*

Line 125: Throughout the text referral to e.g. "Table S1"; the supporting materials appear as Table A1.

*It will be corrected according to the Editorial policy.*

Figure 7: There appears to be a mismatch between the $^{13}$C chemical shifts of COM here and those seen in the 2D HETCOR DNP in Fig. A2

*Good observation indeed. The purpose of Figure A2 is initially to demonstrate to the reader that 4 isotropic peaks are indeed observed in the $^1$H dimension (this is an essential point for the 1H discussion). The following sentence will be added in the final MS (caption of Figure A2):*

*"NB: the temperature used (100 K) has an impact on the values of the $^{13}$C chemical shifts. The main goal is to demonstrate that four $^1$H resonances are clearly observed in the indirect dimension."*

Line 323: " ~ 0.8% of the whole $^{13}$C isotropic chemical shift range " I am not sure what is meant by this statement.

*I agree that this sentence is confusing and not very informative in itself. It will be suppressed.*

Line 325: " evidenced and could be quantified if necessary (by increasing the signal-to-noise ratio significantly) " The S/N seems adequate for coarse quantification which I suggest to include and briefly discuss.

*You are right. The following sentence has been added in the final version:*

*"The S/N is adequate here for coarse quantification. For better accuracy, longer experimental time will be necessary. Moreover, NMR experiments will be combined with denoising techniques developed recently by Laurent et al. (Laurent, Bonhomme, 2020)." This new reference has been added in the bibliographic section.*